# BrewerIX enables allelic expression analysis of imprinted and X-linked genes from bulk and single-cell transcriptomes

Paolo Martini[1,2,7], Gabriele Sales [1,7], Linda Diamante[3], Valentina Perrera[3,4], Chiara Colantuono[5],
Sara Riccardo[5], Davide Cacchiarelli[5,6], Chiara Romualdi [1✉] & Graziano Martello [1✉]

Genomic imprinting and X chromosome inactivation (XCI) are two prototypical epigenetic mechanisms whereby a set of genes is expressed mono-allelically in order to fine-tune their expression levels. Defects in genomic imprinting have been observed in several neurodevelopmental disorders, in a wide range of tumours and in induced pluripotent stem cells (iPSCs). Single Nucleotide Variants (SNVs) are readily detectable by RNA-sequencing allowing the determination of whether imprinted or X-linked genes are aberrantly expressed from both alleles, although standardised analysis methods are still missing. We have developed a tool, named BrewerIX, that provides comprehensive information about the allelic expression of a large, manually-curated set of imprinted and X-linked genes. BrewerIX does not require programming skills, runs on a standard personal computer, and can analyze both bulk and single-cell transcriptomes of human and mouse cells directly from raw sequencing data. BrewerIX confirmed previous observations regarding the bi-allelic expression of some imprinted genes in naive pluripotent cells and extended them to preimplantation embryos. BrewerIX also identified misregulated imprinted genes in breast cancer cells and in human organoids and identified genes escaping XCI in human somatic cells. We believe BrewerIX will be useful for the study of genomic imprinting and XCI during development and reprogramming, and for detecting aberrations in cancer, iPSCs and organoids. Due to its ease of use to non-computational biologists, its implementation could become standard practice during sample assessment, thus raising the robustness and reproducibility of future studies.

[1] Department of Biology, University of Padova, Padua, Italy. [2] Department of Molecular and Translational Medicine, University of Brescia, Brescia, Italy. [3] Department of Molecular Medicine, Medical School, University of Padova, Padua, Italy. [4] International School for Advanced Studies (SISSA/ISAS), Trieste 34136, Italy. [5] Telethon Institute of Genetics and Medicine (TIGEM), Armenise/Harvard Laboratory of Integrative Genomics, Pozzuoli, Italy. [6] Department of Translational Medicine, University of Naples "Federico II", Naples, Italy. [7] These authors contributed equally: Paolo Martini, Gabriele Sales. ✉email: chiara.romualdi@unipd.it; graziano.martello@unipd.it

Gene imprinting is used to control the dosage of a specific set of genes (imprinted genes) by selectively silencing one of the two copies of the gene (either the maternal or the paternal allele). In female cells, also the genes on the X chromosome are expressed mono-allelically thanks to a random epigenetic silencing mechanism called X chromosome inactivation (XCI). However, in humans, a considerable proportion (~20–25%) of genes escape XCI and are expressed bi-allelically.

X-linked and imprinting diseases are the most common congenital human disorders because loss-of-function mutations in the single expressed allele will not be buffered by the second silenced allele[1]. However, thanks to the randomness of XCI, female individuals are often protected from X-linked mutations. Imprinted genes were initially isolated as regulators of fetal growth and their aberrant expression has been related to cancer[2–4]. For these reasons, analyzing the imprinting and XCI status is crucial in many fields, including cancer research, regenerative medicine, and assisted reproductive technology.

Correct imprinting information is used to evaluate the quality of human-induced pluripotent stem cells (iPSCs)[5,6], while reactivation of X chromosome is expected in both human naive and murine naive pluripotent cells[6,7]. Although iPSCs hold the promise for effective approaches in regenerative medicine, disease modeling, and drug screening (for review see Perrera and Martello[6]), their safety is compromised by frequent genetic and epigenetic aberrations, such as Loss of Imprinting (LOI) or a variable X chromosome status[5,8–16]. Organoids are becoming the system of choice for the study of tissue morphogenesis, cancer, and infections[17–21]. However, little is known about their epigenetic stability and, in the case of brain organoids which are commonly derived from PSCs[19–21], it is not known whether epigenetic aberrations found in PSCs[5,6] might be inherited in the organoids.

Although challenging, allelic expression can be determined by the presence of Single Nucleotide Variants (SNVs) in RNA-sequencing (RNA-seq) data. Most of the methods proposed for this aim are based on allelic ratio (AR) thresholding or binomial test[5,22–26], others are specifically designed for reciprocal cross or replicated experimental settings[27–29]. A large scientific literature exists on this issue, sometimes giving contradictory results[24–26,30–32], highlighting the difficulty of this type of analysis. However, at the time of writing, no standardized pipelines for analysis of allelic expression of imprinted and X-linked genes have been developed. Existing pipelines use different combinations of tools and rely on different parameters that were set to analyze specific data and to address specific questions[5,33,34]. Moreover, these pipelines need skilled bioinformaticians to be run. A complete and easy-to-use tool, which does not require programming skills, is still missing.

Motivated by this need, we built BrewerIX, an app available for macOS and Linux Systems that looks for biallelic expression of experimentally validated imprinted genes (see Supplementary Data 1 for a manually curated list of human and mouse genes) and genes on the sex chromosomes. Biallelic expression of imprinted genes will indicate LOI. Biallelic expression of X-linked genes in single cells may indicate reactivation of the X chromosome, as expected in the early embryo[35] or in naive pluripotent stem cells[7,14,15,36], X chromosome erosion, as observed after extensive culture of pluripotent cells[37], or simply escape of single genes from the XCI mechanisms, as recently documented in somatic cells[38,39].

Here we present BrewerIX, a standardized approach for the analysis of known imprinted and X-linked genes. Differently from other tools, its implementation strategy allows the user (i) to perform fast and efficient analyses from raw data to the final plots on a standard desktop or laptop computer without requiring any programming skills, (ii) to have comprehensive information of imprinted and X-linked genes taken from different databases that have been manually curated to avoid results misinterpretation, (iii) to graphically visualize the results in an easy and intuitive way. All these features will guarantee the reproducibility of results and transparency.

## Results

BrewerIX (freely available at https://brewerix.bio.unipd.it) is implemented as a native graphical application for Linux and macOS. It takes as input either bulk or single-cell RNA-seq data (fastq files), analyzes reads mapped over the SNVs distributed on imprinted genes (see "Knowledge base" section for details), X chromosome and Y chromosome, and generates imprinting and XCI profiles of each sample.

BrewerIX implements three pipelines with different aims (Fig. 1a and Supplementary Fig. 1). The Standard pipeline is meant to rapidly have the imprinting and X-inactivation status of a set of samples (Fig. 1a). Here, BrewerIX will align each sample, filter alignments, and call Allele-Specific Expression (ASE) Read counter (see sections below for technical details) using a set of pre-compiled biallelic SNVs. Before visualization, SNVs are collapsed by genes to create a table that is displayed by the user interface (UI). The Complete pipeline sacrifices speed for the sake of completeness by using a larger set of SNVs (the biallelic set used in the Standard pipeline plus the biallelic set called on the user dataset using a pre-compiled set of multi-allelic SNVs). The use of a larger set of SNVs will increase the power to detect biallelic expression. The Tailored pipeline uses a specific set of SNVs that the user might detect from whole-genome or whole-exome sequencing data, allowing to evaluate imprinting and X-inactivation starting directly from the actual SNV profile of the samples (Supplementary Fig. 1). While the input files for the Standard and the Complete pipelines are only fastq files derived from RNA-seq experiments, the Tailored pipeline additionally requires the VCF file with a set of biallelic SNVs. To speed up the analysis BrewerIX allows multicore processing.

The end-point of the pipelines is a table (called "brewer-table") that is visualized by the UI. The UI presents the results using two graphical panels. The gene summary panel shows a matrix of dots with as many rows as the number of genes (ordered according to their genomic position) and as many columns as the number of samples analyzed (position of the samples can be arranged just dragging them in the desired order). The size and the color of the dot are proportional to the confidence of our estimate: (i) the larger the dot, the higher the number of SNVs supporting our estimate; (ii) the brighter the color, the closer to 1 is the average of the allelic ratios (minor/major) of all biallelic SNVs. Empty dots are expressed genes with no evidence of biallelic expression. Gray squares mean that the gene was detected but did not reach the user's thresholds, while the absence of any symbol indicates that the gene was not detected (0 reads mapping on SNVs).

The SNVs summary panel shows a set of barplots (one set for each sample) with as many bars as the number of SNVs per gene. Here, blue is the color of the reference allele and red is the alternative/minor one. Solid colors indicate biallelic SNVs, transparent colors indicate monoallelic SNVs, while those SNVs that do not meet the minimum coverage are shown in gray. When a gene shows no evidence of any genuine biallelic SNVs, we collapse the counts over a virtual SNV (named "rs_multi") to give an indication of its expression.

The UI allows setting different filters: on SNVs and on genes. The filters on SNVs are based on the following four parameters:

1. the overall depth (OD), representing the number of reads mapping on a given SNV;

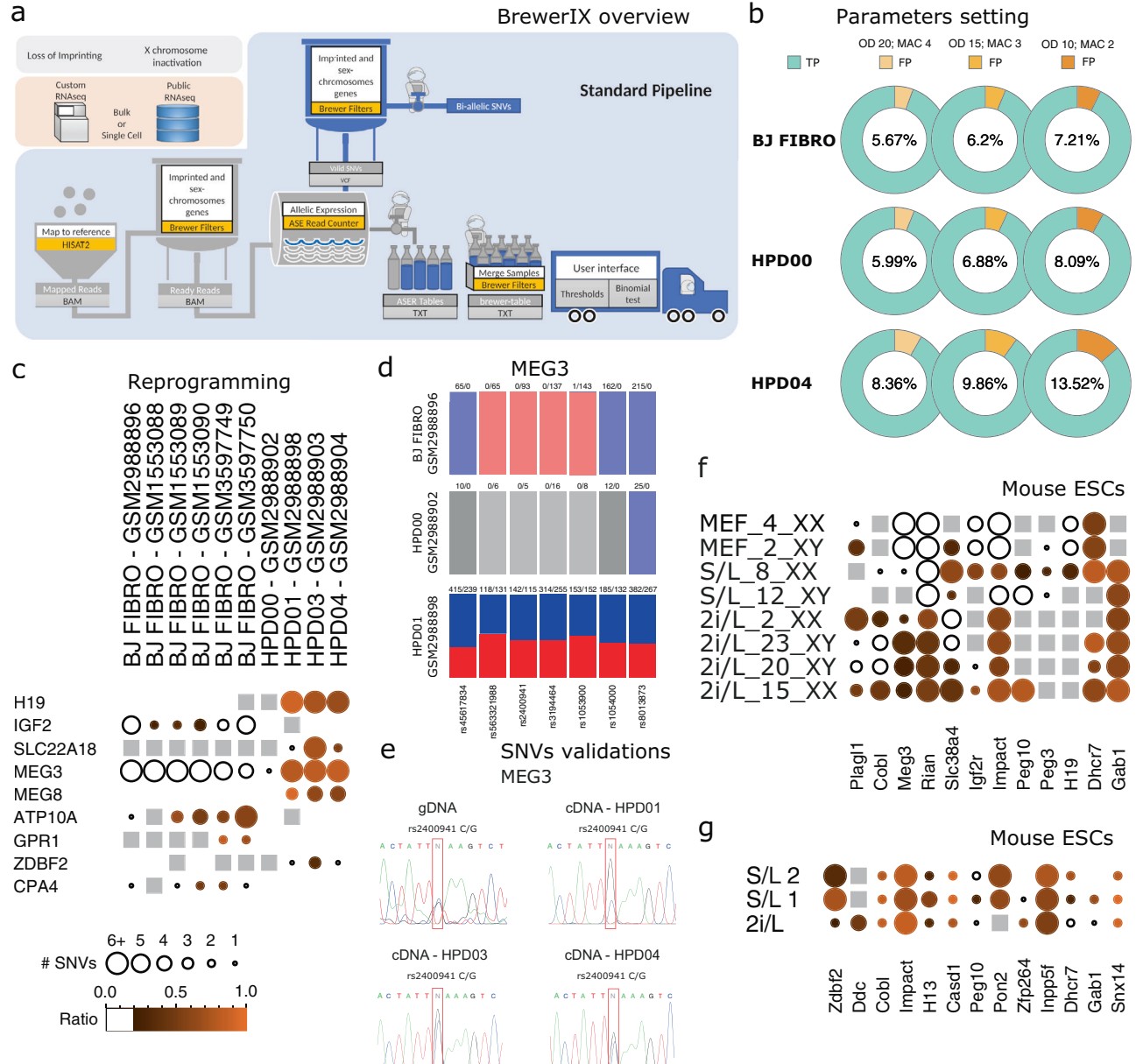

**Fig. 1 Analyses of imprinted gene expression in naive pluripotent cells with BrewerIX. a** BrewerIX rational and overall implementation scheme for the Standard pipeline. **b** False discovery rate estimates obtained by comparing WES calls and BrewerIX biallelic calls in one male BJ fibroblast and two iPS cell lines. Three threshold combinations of overall depth (OD) and minimal coverage of the minor allele (MAC) were used; true positives (TP) in cyan; false positives (FP) in shades of orange. **c** BrewerIX gene summary panel results on bulk RNA-seq data from isogenic human fibroblasts (BJ FIBRO), primed (HPD00) and naive (HPD01/3/4) iPSCs. The larger the dot, the higher the number of SNVs supporting the biallelic call. The brighter the orange, the closer to 1 is the average of the allelic ratios (minor/major) of all the biallelic SNVs. Empty dots indicate detected genes with no evidence of biallelic expression, gray squares indicate genes detected but not reaching the user's thresholds, while the absence of any symbol indicates that the gene was not detected. **d** BrewerIX SNV summary panel for *MEG3* in the case study shown in panel **c**. A barplot for each sample is reported, with as many bars as the number of SNVs per gene. Solid colors represent actual SNV with both loci expressed, blue and red are the reference and the alternative/minor allele. Transparent colors indicate SNVs detected with no evidence of biallelic expression, while grayscale colors indicate SNVs that do not meet the minimum coverage. **e** Experimental validation of the indicated *MEG3* SNVs by PCR followed by Sanger sequencing. The SNVs of interest are highlighted by a red box. See Supplementary Table 2 for a list of all SNVs validated. Each SNVs was detected in two independent experiments, using either forward or reverse sequencing primers. **f** BrewerIX gene summary panel results on bulk RNA-seq data generated by Yagi et al.[47]. Murine ESCs were expanded in either 2i/L or S/L conditions, while mouse embryonic fibroblasts (MEF) serve as controls. **g** BrewerIX gene summary panel results from bulk RNA-seq data of mESCs cultured in 2i/L or S/L (two biological replicates) by Kolodziejczyk and colleagues[48]. See Fig. 2a for matching single-cell RNA-seq samples.

2. the minor allele count (MAC), indicating the absolute number of reads mapping on the less frequent SNV variant among the two detected (i.e., the minor allele);

3. the threshold to call a biallelic SNV, which can be either a cutoff on the allelic ratio (AR, minor/major allele) or the p-value of a binomial test[22,23];

4. the minimal number of biallelic SNVs needed to call a biallelic gene, based on the assumption that when a gene is expressed bi-allelically, multiple biallelic SNVs should be detected.

The filter on genes allows the user to choose the set of genes to display: all, only those detected (i.e., those with a sufficient OD) or only those genes that are biallelic in at least one sample that was analyzed. In addition, the user can control the source of imprinted genes to be included in the analysis: human and mouse have four sources that can be combined (see "Knowledge base" paragraph) with the additional possibility to exclude placental and/or isoform-dependent genes. Finally, the user can control the allelic ratio measure, as either minor allele/major allele, or minor allele/total counts.

Both genes and SNVs summary panels can be saved as PDF files. Moreover, the gene summary panel can be exported as a tab-delimited file to allow further analysis. All exports reflect the filters chosen.

**Default parameters setting**. Default values of the parameters have been empirically selected to minimize the number of false positives. A false-positive call is an SNV not present in the DNA, detected only at the RNA level due to sequencing and aligner errors.

To precisely estimate the rate of false positives (FDR), we generated whole-exome sequencing (WES) data for human male BJ fibroblasts and for two iPS cells lines (HPD00 and HDP04[15]) and identified SNVs on all autosomes of each cell line. We then analyzed with BrewerIX RNA-seq data from the three cell lines and estimated the fraction of SNVs detected from transcriptomic data but not confirmed by WES using different thresholds. As shown in Fig. 1b, we obtained a mean FDR of 6.67% using an AR >= 0.2—as in ref. [5] and[35]—, an OD = 20 reads and MAC = 4. Lowering such parameters to OD = 15 and MAC = 3 did not increase dramatically (7.64%) the FDR in all samples tested (BJ prop test $P$-value 0.4587; HPD00 prop test $P$-value 0.1713; HPD04 prop test $P$-value 0.01829), while a further reduction increased the FDR to 9.60% (BJ prop test $P$-value 0.1086; HPD00 prop test $P$-value 0.04211; HPD04 prop test $P$-value 4.582e-09).

To further evaluate the false-positive calls, we analyzed genes on sex chromosomes of male cells, of whom only a single allele is present. Thus, we analyzed bulk RNA-seq samples of six normal male BJ fibroblasts from three published datasets (see Supplementary Table 1, describing all datasets used in this study). We collected on sex chromosomes all the SNVs with an OD >= 5 reads in at least one sample.

As shown in Supplementary Fig. 2, the mean frequency distribution of false-positive calls on the X chromosome dropped to 3 every $10^5$ SNVs analyzed, using an OD = 15 and MAC = 3. Importantly, no biallelic SNVs were detected on the Y chromosome in any of the analyzed samples.

To gain further confidence in methods based on RNA-seq data, we calculated the number of false-positive calls detected by SNP-array, a technique specifically developed and extensively used to detect SNVs. We analyzed genomic DNA from BJ fibroblasts profiled with Affymetrix Mapping 250 K Nsp SNP Array (GEO accession GSE72531[40]), and we found that the number of false positives detected was 100 times higher (2 every $10^3$ evaluated SNVs, Supplementary Fig. 3) confirming that RNA-seq data is more accurate in detecting allelic imbalance.

Although the defined thresholds minimize false positives, we investigated their power of detecting actual biallelic genes. For this reason, we analyzed RNAs-seq data from female human naive iPSCs (HPD08 - GSM2988908), bearing two active X

chromosomes[15,21]. We detected 104 biallelic genes on the entire X chromosome out of 382 detected genes (27.2%).

We performed a similar analysis on all genes located on autosomes—obviously excluding imprinted genes—in three different cell lines and found that on average 35.5% of the protein-coding genes detected were biallelic (BJ fibroblasts 33%—1145/3471; HPD00 31.5%—1281/4068 and HPD04 42%—1805/4295). Detection of ~30% of biallelic expression from autosomes or from X chromosomes of female naive cells is an expected result, given that not every single transcript might have SNVs allowing the detection of allelic imbalance (see "Discussion") and given that monoallelic or allele-biased expression is observed out of imprinted and X-linked loci[41–43].

Overall, we conclude that the chosen parameters allow the detection of biallelic expression while minimizing false-positive calls.

Our default parameters for standard bulk RNA-seq samples (>10 M reads/sample) are 20, 4, and 0.2 for OD, MAC, and AR, respectively. In addition, we call a gene biallelic when at least two biallelic SNVs are detected, in order to filter out potential sequencing artefacts. To test BrewerIX functionalities we analyzed very diverse datasets, including both bulk and single-cell RNAseq, different organisms (human and mouse) and different biological systems (iPSCs, cancer cells, early embryonic development and organoids).

**Human-induced pluripotent stem cells (iPSCs) and murine embryonic stem cells (mESCs).** Reprogramming of human somatic cells to pluripotency has been associated with imprinting abnormalities[6], both in the case of conventional, or "primed", iPSCs and in the case of naive iPSCs[5,8–10,15,44–46].

We analyzed ten isogenic bulk RNA-seq samples, including six BJ fibroblast, one primed iPSC, and three naive iPSC lines. We run the analysis both with the Complete pipeline (Fig. 1c) and the Standard pipeline (Supplementary Fig. 4), obtaining highly comparable results. *MEG3*, *H19*, and *MEG8* showed biallelic expression specifically in naive iPSCs (Fig. 1c, d), as previously reported[15,46].

To experimentally validate these results and further demonstrate the accuracy of the default parameters, we performed Sanger sequencing after PCR amplification of genomic DNA from 1 naive iPSC line and confirmed the presence of 12 randomly selected SNVs (Supplementary Table 2 and Fig. 1e), while biallelic expression of *MEG3* was confirmed in 3 independent naive iPSC lines (Fig. 1e). An additional dataset of human fibroblasts (HFF) and matching naive iPSCs (HPD06[15]) was analyzed with the Standard pipeline, confirming biallelic expression of *H19* and *MEG3* only in naive cells (Supplementary Fig. 5), as previously reported[15,36,46]. We analyze a dataset of murine Embryonic Stem cells (mESCs) expanded under different culture conditions. Yagi et al. reported that expanding mESCs in 2i/L conditions resulted in LOI, while mESCs in S/L conditions mostly retained correct imprinting[47]. With BrewerIX we obtained highly similar results for the imprinted genes analyzed by Yagi et al. (Fig. 1f) and detected five additional biallelic transcripts. We conclude that BrewerIX detected LOI events in both human and mouse naive pluripotent stem cells from bulk RNA-seq data, in agreement with the previous analyses[15,46,47].

**Single-cell analysis of pluripotent, embryonic, somatic, and cancer cells.** Next, we wanted to compare the performance of BrewerIX on matching bulk and single-cell RNA-seq data. Using bulk samples from mESCs cultured in 2i/L or S/L conditions[48], we identified 13 LOI events, with *Ddc* and *Zfp264* showing LOI

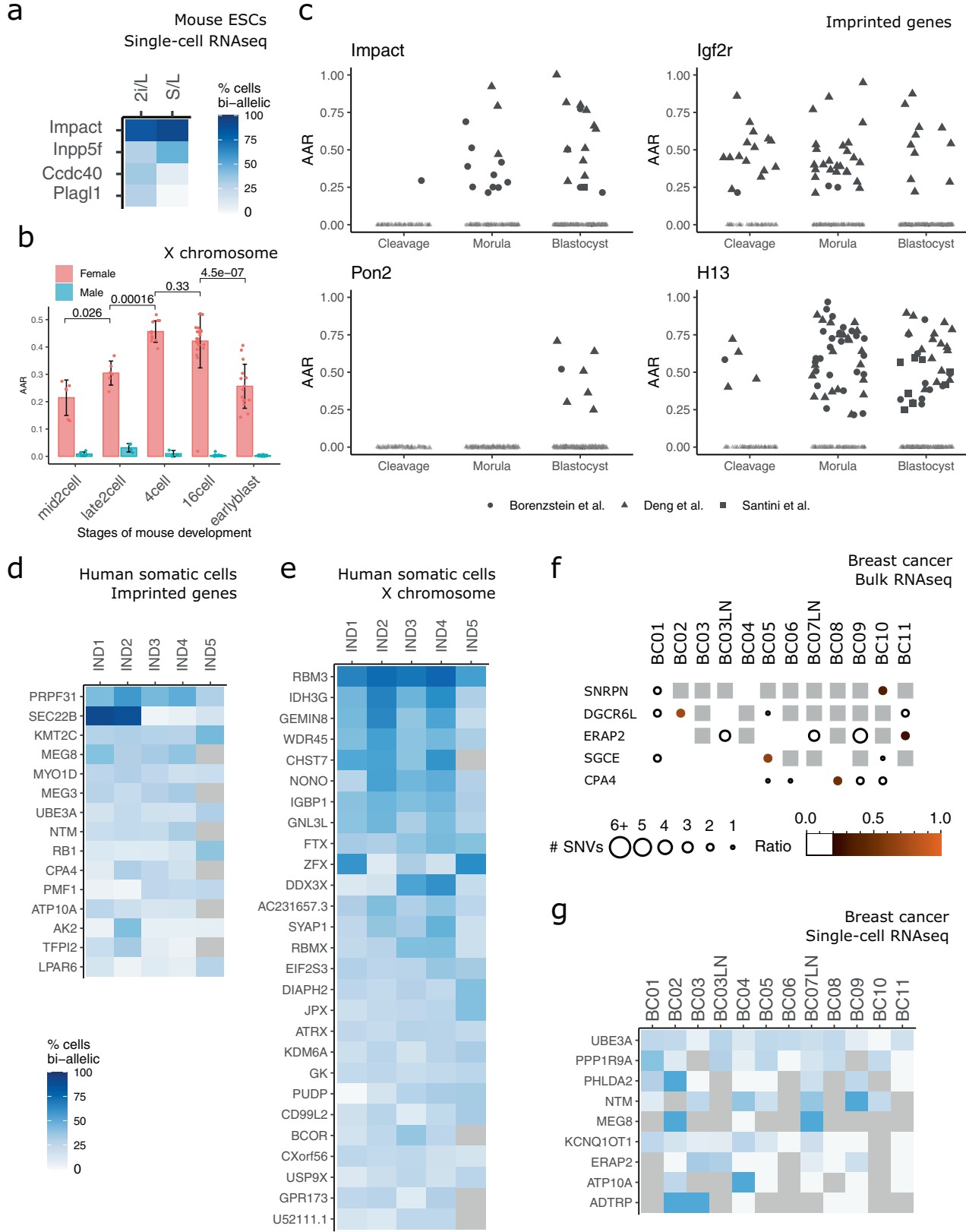

specifically in 2i/L and *Pon2*, *Peg10*, *Dhcr7*, and *Gab1* showing LOI only in S/L (Fig. 1g), and the remaining 7 shared between the two conditions.

We then analyzed single-cell data (384 cells from 2i/L and 288 from S/L) using 15, 3, and 0.2 for OD, MAC, and AR,

respectively, in order to account for the sequencing depth, lower than bulk samples. We also considered a gene bi-allelically expressed when a single SNVs was found biallelic in at least 20% of cells analyzed expressing such gene (Fig. 2a). We observed that *Impact* and *Inpp5*, which displayed multiple biallelic SNVs in

**Fig. 2 Analyses of single-cell RNA-seq data of mouse embryonic and human adult cells. a** Analysis of single-cell RNA-seq data from mESCs cultured in 2i/L or S/L, matching those shown in Fig. 1g. Results are summarized as percentages (degree of blue) of cells in which a given gene was expressed bi-allelically. The number of cells analyzed: 2i/L 384, S/L 288. **b** Average allelic ratio (AAR) is defined as the average of paternal/maternal ratios across single cells for all genes in X chromosome in male and female embryonic cells detected by single-cell RNAseq[41]. Wilcoxon tests were performed between pairs of sequential developmental stages of female embryos (mid2cell—late2cell, late2cell—4-cell, 4-cell– 16cell, 16cell—earlyblast. The number of cells for male (M) and female (F) for each developmental stage: mid2cell 6 M, 6 F; late2cell 4 M, 6 F; 4-cell 3 M, 11 F; 16cell 27 M, 23 F; earlyblast 28 M, 15 F. See also Supplementary Fig. 6. **c** Genes with frequent LOI across mouse developmental stages obtained by studying three datasets[41, 49, 50]. On the y axis, the average allelic ratios (AAR) of single samples (single cells or single embryos for the Santini dataset). Developmental stages have been collapsed into broader categories (cleavage, morula, and blastocyst, see "Methods"). Number of cells for developmental stage: Deng et al. zygote 4, early2cell 8, mid2cell 12, late2cell 10, 4-cell 14, 8-cell 28, 16cell 50, earlyblast 43, midblast 60, lateblast 30; Borensztein et al., 2-cell 6, 4-cell 10, 8-cell 29, 16-cell 15, 32-cell 26, 64-cell 20; Santini et al. Blastocyst 8. See also Supplementary Figs. 7 and 8. **d** Analysis of single-cell RNA-seq data[34] from 772 human fibroblasts and 48 lymphoblastoid cells from 5 female individuals (IND1-5). Results are summarized as percentages (degree of blue) of cells in which a given gene was expressed bi-allelically. Gray indicates undetected genes. Number cells: IND1 229, IND2 159, IND3 192, IND4 192, and IND5 48. **e** Results for X chromosome genes on samples described in panel **d. f** BrewerIX gene summary panel results from bulk RNA-seq data from human breast cancer samples[53]. LN indicates matching metastatic lymph nodes. **g** Analysis of single-cell RNA-seq data from breast cancer samples, matching those analyzed in panel f. Number of cells: BC01 22, BC02 53, BC03 33, BC03LN 53, BC04 55, BC05 76, BC06 18, BC07LN 52, BC08 22, BC09 55, BC10 15, BC11 11. Gray indicates undetected genes.

bulk analysis (Fig. 1g) were found biallelic also in a large fraction (>50%) of single cells analyzed (Fig. 2a). Several LOI events were detected only in bulk samples, possibly because single-cell RNAseq detects preferentially the 3' end of transcripts, limiting the number of SNVs detected. Despite such limitations, some biallelic genes could be detected only by single-cell RNAseq (*Ccdc40* and *Plagl1*), indicating that only single-cell RNAseq allows the detection of LOI events occurring in a limited fraction of cells.

Deng et al. analyzed the gene expression of single cells from oocyte to blastocyst stages of mouse preimplantation development describing that in female embryos the paternal X chromosome is transiently activated at the four-cell stage and subsequently silenced[41]. BrewerIX results were highly concordant with those generated with a custom pipeline by Deng et al., confirming the transient reactivation of the paternal X chromosome (Fig. 2b and Supplementary Fig. 6). Next, we observed an expected monoallelic expression of imprinted genes (Fig. 2c and Supplementary Fig. 7), although 9 of them showed biallelic expression at several stages of preimplantation embryos. We analyzed two additional datasets[49,50] of early mouse embryos and confirmed biallelic expression of such genes in multiple samples from at least two independent studies. Of note, the biallelic expression of *Zim3* and *Usp29* was detected up to the 4-cell stage and could be attributed to a mix of maternal and zygotic mRNAs, each expressing a different allele (Supplementary Fig. 8). Conversely, the remaining seven imprinted genes were biallelic at the blastocyst stage, indicating defective imprinted gene expression.

Next, we analyzed a human somatic single-cell RNA-seq dataset[34] and observed that 15 genes showed biallelic expression in 20% of cells (Fig. 2d). Only two of these genes (*ATP10A* and *TFPI2*) were also found biallelic by the authors of the original study[34]. We extended the analysis to X-linked genes and found that, out of 583 detected genes, 27 genes escaped XCI in at least two individuals (Fig. 2e). Notably, 18 out of 27 were previously identified as escapees[51], while the remaining 9 were identified by BrewerIX. We conclude that BrewerIX efficiently identifies LOI and XCI escape events occurring in small fractions of somatic cells from single-cell transcriptomes.

Different cancers, such as breast, kidney, and lung, are characterized by frequent expression level changes of imprinted genes, often accompanied by DNA methylation level changes in several imprinted domains, such as *PEG3*[52]. To test whether BrewerIX could detect LOI events in cancer cells, we analyzed 515 single-cell samples and matching bulk samples from 11 breast cancer patients[53]. Analysis of bulk samples using the Complete

pipeline identified only five genes, each expressed bi-allelically in only one sample. In stark contrast, analysis of single-cell data identified nine genes biallelic in the majority of breast cancer samples (Fig. 2f).

Such results indicate that single-cell analyses outperform bulk analyses in the case of heterogeneous cancer samples and that imprinting abnormalities might be much more widespread in cancer cells than currently thought.

**Analysis of brain organoids**. Human PSCs have been recently shown to have the capacity to self-organize into 3D structures containing different parts of the brain[19–21,54]. Such structures have been named cerebral organoids, or "mini-brains"[20,21,55]. Moreover, it is also possible to obtain more homogeneous structures such as retina or cortical organoids[19,54,56–58].

We first analyzed single-cell transcriptomes from fetal neural cortex[57] and observed biallelic expression of several imprinted genes (Fig. 3a), some of which were previously reported to be biallelic in the brain, such as *PPP1R9A* and *NTM*[31]. We then analyzed transcriptome from minibrains and cortical organoids and observed that the same genes were found biallelic in multiple independent samples (Fig. 3a, b). We conclude that brain organoids faithfully recapitulate the tissue-specific regulation of imprinted genes observed in the brain. Notably, we also observed that some genes that are known to be imprinted in the brain[57], such as *DLK1*, *SMOC1*, and *L3MBTL1*, were bi-allelically expressed in some organoids, indicating LOI events associated with organoid formation. However, *DKL1* and *L3MBTL1* frequently show LOI in hPSCs, therefore such aberrations might be inherited during neural differentiation, as previously reported[5]. Overall, our results indicate aberrant expression of some imprinted genes in organoids, including genes associated with neurodevelopmental defects and cancer[6].

**Precision/recall analysis and parameters setting**. To guide the user in the process of cutoff selection we performed a precision/recall analysis based on the set of genes detected by RNA-seq data analyses and validated by Sanger sequencing both (i) in our study and (ii) in the study by Santini and colleagues[50] obtaining a total of 307 validated SNV of which 49 are true positives (i.e., validated biallelic) and 258 are true negatives (i.e., validated monoallelic expression).

As expected, in Fig. 4a we can appreciate that the precision has an increasing trend from the lower to higher overall depths thresholds. For allelic ratios (AR) between 0.1 and 0.3, we obtained a precision greater than 80% for overall depth cutoff greater or equal to 15 reads.

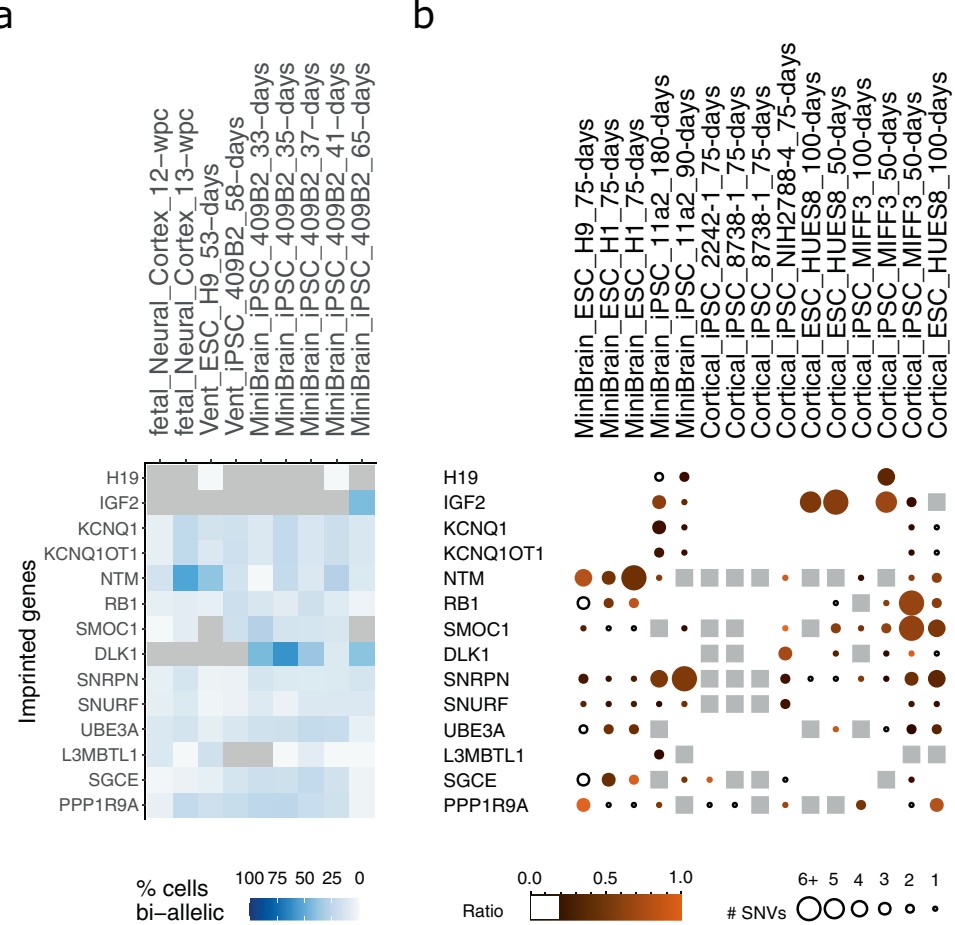

**Fig. 3 Analysis of bulk and single-cell RNA-seq data from human organoids for 14 selected genes. a** Analysis of single-cell RNA-seq data from the fetal neocortex, cortical-like ventricle from cerebral organoids (Vent) and whole-cerebral organoids (minibrains). Gray indicates undetected genes. **b** Summarized view of the imprinting status of 14 selected genes in 4 different studies in human minibrains and cortical organoids.

The recall was above 60% for values equal to, or lower than 20 for the overall depth. In sum, for an AR = 0.2, which is commonly used to define biallelic expression, an OD between 15 and 20 maximizes both precision and recall (green areas in Fig. 4a, b). Of note, the same parameters minimized the FDR calculated from RNAseq and whole-exome sequencing (Fig. 1b).

We also tested the performance of the binomial distribution (Fig. 4c, d), rather than using AR, and found a precision above 90% for P values equal to or below 0.05, regardless of the overall depth, while the recall was affected by the P-value, remaining above 60% only in the case of a P-value equal to 0.05 and an overall depth equal to or below 15.

This analysis, performed on a set of experimentally validated genes, confirms that our default parameter of overall depth = 20, AR = 2 (thus a minor allele = 4) gives the best trade-off for precision and recall. Moreover, using a binomial distribution with a P-value = 0.05 and an overall depth = 15 gives even higher precision with an acceptable recall.

Concluding, BrewerIX identifies with high precision and recall transcripts that are bi-allelically expressed, but such findings need to be validated by independent techniques, such as Sanger sequencing after PCR amplification of the transcripts under analysis (as shown in Fig. 1).

## Discussion

The inference of LOI using RNAseq is surely highly challenging. Although BrewerIX takes into account possible sources of bias such as the presence of duplicated reads and of reads that overlap two SNVs or the quality control of heterozygous sites for ASE, other issues could limit our detection efficiency (e.g., low gene expression, reference bias mappability and the poor mappability of some genomic regions).

Moreover, it is also important to point out that the identification of mono- or biallelic expression by BrewerIX is dependent on the existence of at least one genomic SNV on the locus of interest and it is limited to those genes that are robustly transcribed in the analyzed samples. For these reasons, a gene detected as non-biallelic by the tool is not necessarily referable as monoallelic, since the absence of any evidence of biallelic expression might be due to poor coverage, a negligible expression of the gene or to the absence of detectable SNVs in the locus of interest. This limitation particularly affects the analysis of mono- or biallelic expression in samples derived from inbred mouse strains, where genomic SNVs are extremely rare.

We should also mention that for the study of X-linked genes single-cell RNA-seq data should be used. In normal human female tissue samples, X-inactivation is random, i.e., about 50% of the cells express the paternal and 50% the maternal copy of X, and hence in bulk RNAseq most of the X-chromosomal sites are expressed bi-allelically irrespective of their X-inactivation status. The use of single-cell data (see Fig. 2) circumvents such limitations.

While considering all these limitations, we notice that the results obtained by BrewerIX on the selected case studies out-

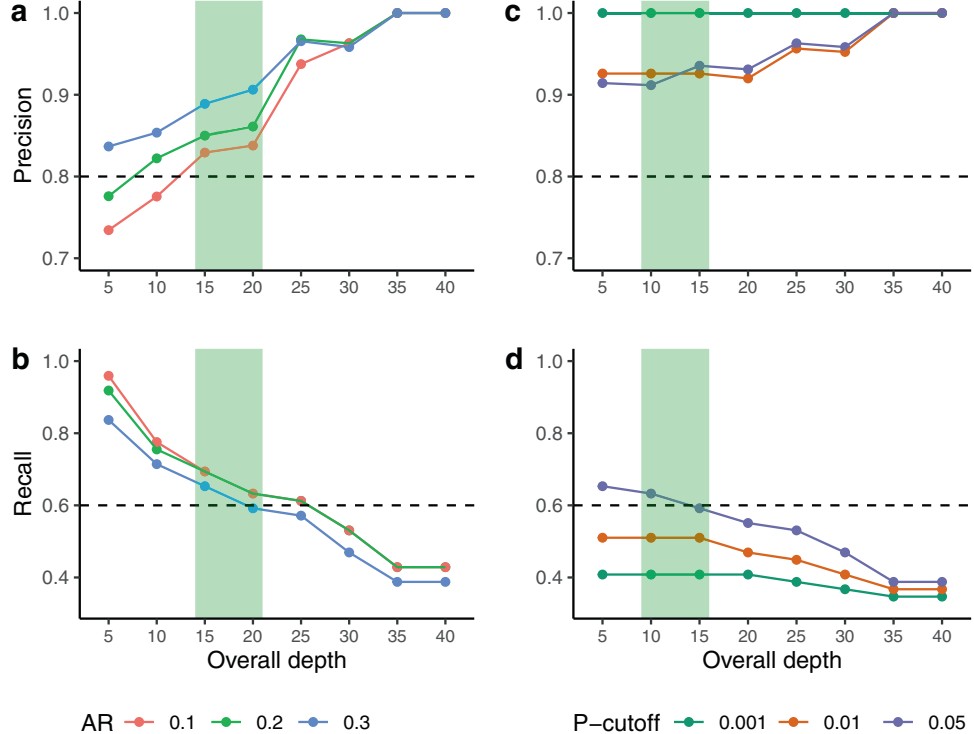

**Fig. 4 Precision-recall analysis.** Precision and recall analysis on the validated SNV from this study and from Santini et al. (**a**) precision and (**b**) recall using increasing Overall Depth (OD) and three different Allelic Ratio (AR). **c** Precision and **d** recall using increasing OD and three *P*-value cutoffs for the binomial test. The horizontal dashed lines define the cutoffs of acceptable precision and recall values, while the green areas indicate the best overall depths.

competed published custom pipelines confirming and extending published results, demonstrating the reliability and usefulness of the tool. For the analysis of relatively homogeneous cell populations, such as pluripotent cells in culture, we conclude that bulk RNA-seq data allowed robust identification of LOI events. Conversely, when heterogeneous populations of cells, such as cancer samples, are analyzed, only single-cell measurements allowed the detection of widespread events of LOI or XCI escape, indicating that such phenomena might have been underestimated for technical limitations.

Previous studies reported biallelic expression of some imprinted genes in both human and mouse naive pluripotent cells[5,46,47] and interpreted it as aberrations induced by in vitro culture under specific conditions. We confirmed such observations for both human and murine cells, the latter showing biallelic expression of several genes regardless of the culture conditions used. Interestingly, analysis of three independent preimplantation embryo datasets showed biallelic expression of multiple imprinted genes, some of which (*H13*, *Impact*, *Dhcr7*, *Snx14*, *Igf2r*, and *Pon2*) were also biallelic in mESCs. Of note, similar conclusions have been drawn by Santini and colleagues[50], suggesting that biallelic expression of some imprinted genes is normally occurring in naive pluripotent cells in vivo and in vitro.

Due to the ease of use of BrewerIX to noncomputational biologists, we believe that its implementation could become standard practice during the assessment of newly generated pluripotent cells and organoids, as well as for the study of the molecular mechanisms underlying genomic imprinting and XCI in different tissues and developmental stages, hopefully raising robustness and reproducibility of future studies.

## Methods

### Knowledge base
. The knowledge base contains the species genome with the genome index (for hisat2), the biallelic and the multi-allelic SNV file (ENSEMBL

variants annotation version 98; INDELs and the SNVs whose reference alleles differ from the reference genome were removed), the regions with the genes of interest i.e., imprinted genes and genes on the sex chromosomes.

We manually curated a comprehensive set of imprinted genes from different sources. For human and mouse imprinted genes, we collected the data from the Geneimprint database (http://geneimprint.com/) and Otago database (http://igc.otago.ac.nz/home.html). We excluded all genes labeled as "Predicted" or "Not Imprinted" and manually curated "Conflicting Data". We added human imprinted genes identified by Santoni et al.[34] and mouse imprinted genes regulated by H3K27me3 in the early embryo, identified by Inoue et al.[59]. We have also labeled placental-specific and isoform-dependent imprinted genes within the curated gene list. Placental-specific imprinted genes were identified by combining information from the Otago database and from two additional studies—[60] for human imprinted genes and[61] for mouse genes. For isoform-specific genes, we referred to Geneimprint (category: "Isoform Dependent") and Otago databases. The manually curated gene lists are shown in Supplementary Data 1.

**Front-end implementation.** The BrewerIX graphical interface is distributed as a native application for both Linux and macOS. It is written in the Haskell programming language and makes use of the wxWidgets cross-platform GUI library. Plots are generated using the Cairo library and its PDF output capabilities. The Linux version of the application is packaged using the AppImage tool.

**Back-end implementation.** The computational pipeline is implemented in Python and is available as a Python package called brewerix-cli at https://github.com/Romualdi-Lab/brewerix-cli. The pipeline performs the alignment, allelic count and creation of the "brewer-table". The pipeline can be run also using the command line interface (CLI) implemented by brewerix-cli itself. The final output of the CLI is the "brewer-table" that is parsed by the user interface to produce the BrewerIX visual outputs. The CLI has been thought for advanced users willing to analyze their own set of genes or genomes of different species. The minimum required inputs are the following: a genome (fasta format) and its index for hisat2, genome dict (computed with GATK) and genome fasta index, a bed file indicating the region of interest (i.e., imprinted genes and genes on the sex chromosomes), a set of biallelic SNVs with reference alleles that must be present in the reference genome. In the following, we report the technical details of each analysis step.

*Alignments.* BrewerIX requires fastq files as input. The pipeline works with a homogeneous library layout i.e., all fastq files are either single- or paired-end. The fastq files are aligned to a reference genome. The user can choose between Mouse GRCm38.p6 or human GRCh38.p13 genome. Alignments are performed using

hisat2 (version 2.1.0, default parameters) and filtered to keep only reads laying on genes of interest. Aligned reads are further processed according to GATK best practices, i.e., marking duplicates, splitting reads with N in the cigar, and performing base quality scores recalibration (such post-processing steps are optional in brewerix-cli).

*SNV calling.* SNVs are called only at multi-allelic SNVs using HaplotypeCaller from GATK v4.1. Calls are performed as if all the samples have the same genotype, i.e., all in the same batch. The reference and the most represented alternative allele are selected. We set the following parameters: "--max-alternate-alleles 1 -stand-call-conf 1 --alleles multi_allele_vcf_file --dbsnp multi_allele_vcf_file".

*Allelic count.* The allelic count is performed using ASEReadCounter with default parameters from GATK v4.1. This tool, given a set of loci and a bam file, allows computing the reads bearing the reference and the alternative allele. Sample-specific results are collapsed into an ASER table.

**WES library preparation**. Genomic DNA was extracted from BJ fibroblasts and two iPS cell lines, HPD00 and HPD04; the human foreskin fibroblast cell lines BJ were cultured in DMEM with 10% fetal bovine serum (FBS; Sigma-Aldrich), in normoxic conditions (21% $O_2$, 5% $CO_2$, 37 °C), while iPSCs were cultured on MEFs in the FGF-free medium RSeT (05970, StemCell Technologies, prepared following the manufacturer's instructions) in hypoxic conditions (5% $O_2$, 5% $CO_2$, 37 °C), as described in ref. [15]. Two replicates for each cell line were used and gDNA was quantified using the Qubit 2.0 fluorimetric Assay (Thermo Fisher Scientific); sample integrity, based on the DIN (DNA integrity number), was assessed using a Genomic DNA ScreenTape assay on TapeStation 4200 (Agilent Technologies).

Libraries were prepared from 100 ng of total DNA using a WES service (Next Generation Diagnostics srl) which included library preparation, target enrichment using the Agilent V7 probe set, quality assessment, and sequencing on a NovaSeq 6000 sequencing system using a paired-end, 300 cycle strategy (2 × 150) (Illumina Inc.).

*WES analysis.* The raw data were analyzed by Next Generation Diagnostics srl Whole Exome Sequencing pipeline (v1.0) which involves a cleaning step by UMI removal, quality filtering and trimming, alignment to the reference genome, removal of duplicate reads, and variant calling[62–65]. Variants were finally annotated by the Ensembl Variant Effect Predictor (VEP) tool5.

The final set of variants was refined applying hard-filter according to GATK best practices. In detail, we used GATK VariantFiltration QD < 2.0, FS > 60.0, MQ < 40.0, SOR > 4.0, MQRankSum < −12.5 and ReadPosRankSum < −8.0.

*False discovery rate.* From the BrewerIX ASER table obtained from the RNA-seq data, we extracted the SNVs with an OD = 5 reads and MAC ≥ 1. Using this set of SNVs we computed ASE from WES data. For each sample, we extracted the set of SNV covered by at least five reads in RNAseq and in both WES replicates. False positives are defined as the number of BrewerIX biallelic SNVs without an heterozygous call in at least one WES replicate. False positives were computed at three thresholds: 20—4; 15—3 and 10—2, respectively, for OD and MAC.

**Precision/recall analysis**. To estimate precision and recall, we build a set of true positive and true negative calls that were experimentally validated both in this study and in[50]. We expanded our previous validations reported in Supplementary Table 2 with five additional SNVs (collecting in total 17 SNVs validated in 5 samples). We downloaded from www.sanger.ac.uk/sanger/Mouse_SnpViewer/rel-1303 the SNVs of the two mouse strains (C57BL/6 [B6] and DBA/2) used by Santini et al. and extract 135 SNVs laying in 11 monoallelic validated genes (see Santini's[50] Supplementary data 3). We summarized the BrewerIX calls for the SNVs validated in this study and those validated by Santini and colleagues in Supplementary Data 2 (only SNV with more than five reads were considered).

**Case studies data**. All RNA-seq data but three were downloaded from GEO database using fastq-dump from sra-tools version 2.8.2. The mouse ESCs, bulk, and single-cell Cortical organoid datasets were downloaded from Array Express via direct link. All datasets images were created using the BrewerIX-core imprinted genes (i.e., genes curated from Geneimprint DB and Otago) unless stated otherwise. Moreover, images were created showing only relevant genes with default parameters i.e OD = 20, MAC = 4, AR > = 0.2 and at least two biallelic SNV per gene in bulk data, while we considered a gene biallelic when at least one SNV was found biallelic with OD = 15 and MAC = 3 in the case of single-cell data. For the organoid dataset, we manually selected a list of 14 imprinted genes for Fig. 3a, b, moreover in the case of droplet-based single-cell RNA-sequencing data we had to consider the samples as pseudo-bulk, rather than single-cell datasets, because the detected SNVs were too few, independently from the sequencing depth.

We collected BJ fibroblasts RNA-seq data from three sources on the GEO database: GSE110377[15] (BJ fibroblast GSM2988896; primed iPSC GSM2988902, naïve iPSC GSM2988898, GSM2988903, GSM2988904), GSE126397[66] (BJ fibroblasts GSM3597749 and GSM3597750) and GSE63577[67] (BJ fibroblasts GSM1553088-GSM1553090). To deal with the heterogeneous reads layout (single-

and paired-end) of the sequencing data, we aligned each batch to the reference human genome using hisat2, with default parameters. We use BrewerIX-cli to run the analysis starting from the alignment files (bam). We used the Complete pipeline and loaded the "brewer-table" on the visual interface to explore the results.

HFF samples were downloaded from GSE93226 (GSM2448850-GSM2448852) while reprogrammed iPSC from GSE110377[15] (GSM2988900). As for the BJ fibroblast dataset, we computed single- and paired-end alignments separately (hisat2, default parameters) and then run brewerix-cli with Standard pipeline. Panels summarizing the results have been generated with BrewerIX user interface.

The Yagi dataset of mESCs (GEO accession GSE84164[47]; GSM2425488-GSM2425495) was fully analyzed by BrewerIX with the Complete pipeline.

For the Kolodziejczyk et al.[48] and Kim et al.[68] dataset of mouse ESCs we analyzed mESCs cultured in 2i/L or S/L downloaded from Array Express under the accession E-MTAB-2600[48,68]. We analyzed three bulk samples (one cultured in 2i/L and two in S/L) and 682 single-cell samples (384 cultured in 2i/L and 288 in S/L). Both bulk and single-cell RNA-seq datasets were analyzed using BrewerIX with Standard pipeline. Bulk data visualization of the three samples was performed using BrewerIX user interface. Single-cell RNA-seq results were visualized using custom R code available at github.com/Romualdi-Lab/. Results were summarized by the two categories: 2i/L and S/L. We analyzed genes that are expressed in at least ten cells in at least one category. We considered a gene bi-allelically expressed when at least one SNV was found biallelic in at least 20% of cells analyzed expressing such gene (other parameters remain default).

The single-cell datasets of mouse embryos, from oocyte to blastocyst from Deng et al.[41], were downloaded from GEO accession GSE45719[41] (GSM1112490-GSM1112581 and GSM1112603-GSM1278045; female samples include GSM1112504-GSM1112514, GSM1112528-GSM1112539, GSM1112543-GSM1112553, GSM1112626-GSM1112640, GSM1112656-GSM1112661, GSM1112696-GSM1112697, GSM1112702-GSM1112705; male samples include GSM1112490-GSM1112503, GSM1112515-GSM1112527, GSM1112540-GSM1112542, GSM1112554-GSM1112581, GSM1112611-GSM1112625, GSM1112641-GSM1112653, GSM1112654-GSM1112655, GSM1112662-GSM1112695, GSM1112698-GSM1112701, GSM1112706-GSM1112765; for remaining samples no sex specification were available). Analysis has been carried out using BrewerIX with Standard pipeline. The computed values were used for downstream custom analysis (code can be found at https://github.com/Romualdi-Lab/).

For the X chromosome, we performed the analysis plotting the average of the allelic ratios in each developmental stage for male and female samples. We used developmental stages where both male and female samples were present. Thus, we considered 4 male, 6 female in middle 2-cell (mid2cell); 4 male, 6 female for late 2-cell (late2cell); 3 male, 11 female for 4-cell (4-cell); 27 male, 23 female for 16-cell (16cell); 28 male, 15 female for early blastocyst (earlyblast). To detect paternal X chromosome reactivation, we inferred that the maternal allele was the most expressed allele in the mid2cell stage. Using maternal and paternal alleles inferred from the mid2cell stage, we computed the maternal/paternal ratio in all other stages. To evaluate the performance of BrewerIX in detecting paternal X chromosome reactivation, we downloaded Deng's processed dataset from the supplementary material of the manuscript[41]. To avoid any bias, we analyzed genes shared by Deng's processed dataset and BrewerIX generated data.

For imprinted genes, we plotted the Average Allelic Ratio (AAR) for each gene in each developmental stage. We grouped the samples into the following 3 categories: Cleavage: early2cell, mid2cell, late2cell, 4-cell; Morula: 8-cell, 16cell; Blastocyst: earlyblast, midblast, lateblast.

For the datasets of mouse embryos from oocytes to blastocysts, generated by Borensztein and colleagues[49] we analyzed Xist-wt single-cell samples from GSE80810[49] (GSM2371473-GSM2371585). We run BrewerIX with the Complete pipeline. We consider a gene biallelic when at least one SNV was found biallelic with OD = 15 and MAC = 3. We grouped samples into the following three categories: Cleavage stage: 2-cell, 4-cell; Morula 8-cell, 16-cell; Blastocyst 32-cell, 64-cell.

For the blastocysts stage embryos dataset from Santini et al.[50], we analyzed eight samples of blastocyst-stage embryos from GSE152106[50] (GSM4603204-GSM4603211). We run BrewerIX with the Complete pipeline. We considered a gene biallelic when at least one SNV was found biallelic with OD = 15 and MAC = 3.

We used available data from 772 human fibroblasts we analyzed 229, 159, 192, and 192 for IND1, IND2, IND3 and IND4 respectively) and 48 lymphoblastoid (IND5) cells from 5 female individuals (GEO accession GSE123028[34,51], GSM3493332-GSM3494151).

The single-cell RNA-seq dataset was analyzed using BrewerIX with the Standard pipeline. The single-cell RNA-seq visual reports were produced with custom R code available at https://github.com/Romualdi-Lab/.

Results were summarized by individuals. We analyzed genes that are expressed in at least ten cells in at least four individuals. For this dataset, we included all human sources of imprinted genes, i.e., genes curated from geneimprint DB, Otago, and from Santoni et al.[34] (see "Knowledge base" paragraph for details). We considered a gene bi-allelically expressed when at least one SNV was found biallelic in at least 20% of analyzed cells that express that gene (other parameters remain default).

Chung and colleagues[53] analyzed 11 patients representing four subtypes of breast cancer (luminal A—BC01 and BC02, luminal B—BC03, HER2 + − BC04, BC05, and BC06 or triple-negative breast cancer—TNBC— BC07-11). They obtained 515 single-cell transcriptome profiles and 12 matched samples with bulk RNAseq from 11 patients (GEO accession GSE75688[53] all the samples listed in

GSE75688_final_sample_information.txt.gz; B03 has both primary breast cancer and lymph node metastases). Bulk samples from the breast cancer dataset were analyzed using BrewerIX with Complete pipeline. Visual inspection was performed using BrewerIX. The single-cell RNA-seq dataset was run using the Complete pipeline. The single-cell RNA-seq visual reports were produced with custom R code.

Patients were included in the analysis if a corresponding bulk sample was analyzed. The number of cells analyzed for each patient are the following: BC01 = 22, BC02 = 53, BC03 = 33, BC03LN = 53, BC04 = 55, BC05 = 76, BC06 = 18, BC07LN = 52, BC08 = 22, BC09 = 55, BC10 = 15 and BC11 = 11. We analyzed genes that were expressed in at least two cells in at least one sample. We considered a gene bi-allelically expressed when at least one SNV was found biallelic in at least 20% of analyzed cells that express that gene (other parameters remain default). Code to reproduce the figure can be found at https://github.com/Romualdi-Lab/ as well.

In their study, Camp et al.[57] analyzed 734 single-cell transcriptomes from human fetal neocortex or human cerebral organoids (GEO accession GSE75140, GSM1957048-GSM1957493, GSM1957495, GSM1957497, GSM1957499, GSM1957501, GSM1957503, GSM1957505, GSM1957507, GSM1957509, GSM1957511, GSM1957513, GSM1957515, GSM1957518, GSM1957520, GSM1957522, GSM1957524, GSM1957526-GSM1957814). We run the analysis using BrewerIX with Standard pipeline. The single-cell RNA-seq visual reports were produced with custom R code starting from the "brewer-table". Single cells were summarized according to their annotation available from GEO: we collapsed according to the tissue of origin ("Dissociated whole cerebral organoid"; "Fetal neocortex" and "Microdissected cortical-like ventricle from cerebral organoid") and the stage (12 weeks post-conception—12-wpc, 13 weeks post-conception 13-wpc, 33 days, 35 days, 37 days, 41 days, 53 days, 58 days, 65 days). The numbers of cell analyzed for each tissue stage combination are the following: fetal_Neural_Cortex 12-wpc = 164, fetal_Neural_Cortex 13-wpc = 62, Vent_ESC_H9 53 days = 96, Vent_iPSC_409B2 58 days = 79, MiniBrain_iPSC_409B2_33-days = 40, MiniBrain_iPSC_409B2 35 days = 68, MiniBrain_iPSC_409B2 37 days = 71, MiniBrain_iPSC_409B2 41 days = 74, MiniBrain_iPSC_409B2 65 days = 80. We analyzed genes that were expressed in at least ten cells in at least one category. We considered a gene bi-allelically expressed when at least one SNV was found biallelic in at least 20% of analyzed cells expressing that gene (other parameters remain default for single-cell data).

Giandomenico et al.[55] profiled three neural organoids derived from H9 and H1 (2) iPSC using 10Xv2 (GEO accession GSE124174). Sequencing data were used as bulk samples, i.e., not dividing in single cells, because the detected SNVs were too few, independently from the sequencing depth. We selected one run per organoids to avoid any depth biases (SRA run ids SRR8368415, SRR8368423, and SRR8368431). The analysis was run using BrewerIX with Complete pipeline, with default parameters.

In their study, Quadrato et al. profiled organoids at 6 and 3 months age using single-cell sequencing (DropSeq; GEO accession GSE86153[21]). Sequencing data were used as bulk samples, i.e., not dividing in single cells, because the detected SNVs were too few, independently from the sequencing depth. We selected one run per organoids to avoid any depth biases (SRA run ids SRR4082002 and SRR4082026). The analysis was run using BrewerIX with the Complete pipeline, with default parameters.

Pasca et al.[58] analyzed the expression of Cortical Organoids (GEO accession GSE112137). We analyze 4 samples (bulk RNAseq) from four control cortical organoids (GSM3058370, GSM3058382, GSM3058394, and GSM3058406). The analysis was run using BrewerIX with the Complete pipeline, with default parameters.

In their study, Lopez-Tobon et al. profiled cortical organoids both in bulk (Array Express accession E-MTAB-8325[56]) and single-cell RNAseq (10Xv2 - Array Express accession E-MTAB-8337[56]). We analyzed single-cell experiments as bulk samples, i.e., not dividing in single cells, because the detected SNVs were too few, independently from the sequencing depth. Overall we analyzed 3 cortical organoids derived from ESC (HUES8, bulk at 50 and 100 days—Array Express run ids ERR4198631 and ERR4198637; single-cell at 100 days—Array Express run id ERR4229837) and 3 cortical organoids derived from iPSC (MIFF3, bulk at 50 and 100 days— Array Express run id ERR4198633 and ERR4198639; single-cell at 50 days—Array Express run id ERR4229861). We run bulk and single-cell RNAseq separately using BrewerIX with default parameters.

**SNV detection via PCR followed by Sanger sequencing**. Genomic DNA (gDNA) was extracted from cellular pellets with Puregene Core Kit A (Qiagen) according to the manufacturer's protocol; 1 μg gDNA was used as a template for PCR using the Phusion High-Fidelity DNA polymerase (NEB, cat. M0530L).

Total RNA was isolated from cellular pellets using a Total RNA Purification kit (Norgen Biotek, cat. 37500), and complementary DNA (cDNA) was generated using M-MLV Reverse Transcriptase (Invitrogen, cat. 28025-013) and dN6 primers (Invitrogen) from 1000 ng of total RNA following the protocols provided by the manufacturers, including a step of TurboDNAse treatment (Thermo Scientific). cDNA was diluted 1:5 in water and used as a template for PCR using the Phusion High-Fidelity DNA polymerase; gDNA and cDNA were amplified by PCR using primers detailed in Supplementary Table 3. PCR was conducted with the following program: denaturation at 98 °C for 30 s; 35 cycles of denaturation at 98 °C for 10 s,

annealing at a temperature depending on primer sequence (Tm–5 °C) for 30 s, elongation at 72 °C for 15 s; final elongation at 72 °C for 10 min.

PCR reaction products were resolved and imaged by agarose gel electrophoresis. The remaining PCR products were purified using the QIAquickPCR purification kit (Qiagen, cat. 28106) and direct sequencing was performed using the same primers used for PCR amplification. Each PCR region of interest was sequenced at least twice, using both forward and reverse primers. Sanger sequencing was performed by Eurofins Genomics (https://www.eurofinsgenomics.eu/en/custom-dna-sequencing/gatc-services/lightrun-tube/). Sequence analysis and peak detection were performed using freely available ApE software (https://jorgensen.biology.utah.edu/wayned/ape/).

**Statistics and reproducibility**. For RNA-seq and WES datasets, we analyzed two independent biological replicates and, when possible, included samples from multiple studies. Biological replicates indicate when a cell line was exposed to a given treatment multiple times and the samples were harvested, processed, and analyzed all at once. All statistics were done using R (v4.1.1) unless otherwise stated. Fig 1b, P values were computed with prop.test (R package stats v4.1.1) two-sided with Yates' continuity correction. For Fig. 2b, P values were computed with Wilcoxon tests (wilcox.test R package stats v4.1.1) two-sided. Error bars indicate + / − one standard deviation. The binomial test implemented in BrewerIX is performed by scipy v1.7.3 Python v3.8 (scipy.stats.binom_test testing proportion greater than 1/6). No P values were computed when $n < 3$.

**Ethics approval**. All animal experiments were performed according to guidelines and ethical considerations, as outlined in[41,47,49]. The use of human primary cells, human pluripotent stem cells and organoids was approved by ethics committees, as outlined in[55,57,58,67] and informed consent was obtained from all participants, as outlined in[34,51,53].

## Data availability
All RNA-seq data used in this study were publicly available and obtained from either the Gene Expression Omnibus (GEO) database under the accession codes GSE110377, GSE126397, GSE63577, GSE93226, GSE84164, GSE123028, GSE45719, GSE75688, GSE75140, GSE124174, GSE86153, GSE112137, GSE80810, and GSE152106 or from Array Express under the accession codes E-MTAB-2600 and E-MTAB-8325. Whole-exome sequencing data generated in the current study are available via the Sequence Read Archive (SRA) repository with BioProject ID PRJNA705070. Additional details about all datasets in the study are in Supplementary Table 1. The raw Sanger sequencing data file underlying Fig 1e and Supplementary Table 2 have been uploaded on figshare (https://doi.org/10.6084/m9.figshare.17313110). Source data underlying the graph and charts presented in the main figures are available in Supplementary Data 3.

## Code availability
BrewerIX is freely available for academic users at https://brewerix.bio.unipd.it. All code and tutorials are available at https://github.com/Romualdi-Lab under AGPL3 license and have been deposited at Zenodo[69–72].

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

## Acknowledgements

The authors thank members of the Martello laboratory for discussions and suggestions. G.M.'s Laboratory is supported by grants from the Giovanni Armenise–Harvard Foundation, the Telethon Foundation (TCP13013), and an ERC Starting Grant (MetEpiStem), C.R. is supported by the Italian Association for Cancer Research (AIRC) [IG21837], P.M. has been supported by the European Molecular Biology Organization (EMBO) Short-Term Fellowship [8517].

## Author contributions

G.M., C.R., and P.M. designed the study; P.M. and G.S. developed the software and all custom code; P.M. performed all analyses; V.P. and L.D. performed validation experiments; D.C., C.C., and S.R. performed WES analyses. P.M. and L.D. prepared the figures; P.M., C.R., and G.M. wrote the manuscript with input from all authors; G.M. and C.R. supervised the study and provided funding.

## Competing interests

The authors declare no competing interests.
