## [Peer Review File · Communications Biology]

Reviewers' comments:

Reviewer #1 (Remarks to the Author):

The manuscript presents a software tool, BrewerIX, for the study of loss-of-imprinting (LOI) and X-inactivation from RNA-seq data coupled with genotype information, designed to serve as a standardised and easy to use pipeline for such analyses. After the first round of review, the manuscript was transferred to Communications Biology and I am now commenting on this manuscript as a new reviewer. While I have not seen the original manuscript, based not the rebuttal and the new version of the manuscript, I find the authors have done comprehensive work in revising the text and analyses. Particularly, the rationale for the development of the tool is now clearly stated.

While I have no major comments, I ask the authors to address a few points:

My biggest concern is the suitability of their software tool for the analysis of X-inactivation using human bulk RNA-seq. In the introduction, the authors mention the "random epigenetic silencing mechanism called X chromosome inactivation (XCI)", yet I failed to find any discussion on how the randomness of the mechanism potentially impacts the use of ASE for the analysis of X-inactivation. In normal human female tissue samples, X-inactivation is indeed random, i.e. about 50% of the cells express the paternal and 50% the maternal copy of X, and hence in bulk RNA-seq most of the X-chromosomal sites are expressed bi-allelically irrespective of their X-inactivation status (i.e. escapee or silenced). In single cell RNA-seq the randomness of X-inactivation is obviously not an issue, and e.g. in mouse X-inactivation is imprinted with preferential inactivation of the paternal X, and therefore the suggested tool should work fine for the inference of X-inactivation in these types of data. While it appears that the authors do not present any X-inactivation results from human bulk RNA-seq data, it should nevertheless be made very clear in the manuscript that such a limitation in the application of the tool exists.

Two sentences in the introduction need clarification / correction:

1. "the genes on the X chromosome are expressed mono-allelically" In female samples, this mono-allelic expression does apply to the majority of the X chromosome genes, but a considerable proportion (~20-25%) of X-linked genes escape from X-inactivation in humans and are expressed bi-allelically. Additionally, in relation to my first comment, in bulk RNA-seq it is unlikely that a large number of X chromosome genes are mono-allelic in a human female sample, unless there is considerable skewing of X-inactivation.
2. "X-linked and imprinting diseases are the most common congenital human disorders because loss-of-function mutations in the single expressed allele will not be buffered by the second silenced allele" While this statement is true for imprinted genes, again I ask the authors to specify how this holds for the X chromosome. Given the randomness of X-inactivation, females are most often protected from X-linked of mutations.

While I appreciate the idea of the tool be as a low threshold option for LOI and chrX analyses, I would call for a more extensive discussion on the potential limitations there are in such an automated pipeline vs. customised settings. These limitations may include, e.g., reference allele bias in RNA-seq alignment, handling of RNA-seq reads that overlap multiple SNVs, quality control of heterozygous sites for ASE, poor mappability of some genomic regions, and handling of duplicated RNA-seq reads, that I did not see were specifically discussed or included in the methods.

Reviewer #2 (Remarks to the Author):

The manuscript by Martini and colleagues reports on BrewerIX, a software aimed at assessing the bi-allelic expression of imprinted and X-linked genes from RNAseq data. It is essentially a GUI to HISAT2 for read mapping and GATK for calling SNVs and computing the allelic ratio at each SNV. BrewerIX calls SNVs, then genes, as bi-allelic or non-bi-allelic based on a predefined threshold on the allelic ratio or on the p-value of a binomial test.

Assessing allelic imbalance in RNAseq data is not easy. The caveats of the different approaches were best illustrated by the controversy about the number and identity of imprinted genes that followed the publication of a manuscript by Gregg and colleagues a decade ago (1). The Babak, Dulac, and Gregg groups then proposed modifications of the existing statistical methods that led to contradictory conclusions (2–6). Other groups also proposed different parametric methods (7–9). More recently, a non-parametric method that calls both bi-allelic and biased expression was proposed (10). Despite sophisticated statistical methods, the use of replicated RNAseq data, and reciprocal crosses of mouse strains, the conflicting results illustrated the difficulty of the task and the caution that should be taken to call a gene expression biased.

Considering the existing literature, the approach proposed by Martini and colleagues is naive and simplistic. It is just not possible to reliably call bi-allelic gene expression by only filtering allelic ratios. The second calling method is to use a binomial test. Given the literature above, this is again a simplistic approach unable to deal with the intrinsic complexity of the task. Furthermore, the manuscript offers no detail on how the test is performed exactly and does not mention multiple testing correction; the reference given to support the use of the test (#29) is a review article on X inactivation, which does not describe binomial test whatsoever.

In their rebuttal to reviewer 2, the authors emphasized the fact that BrewerIX is easy to install, fast, and does not require computing skills. This is likely true, but clearly at the expense of accuracy and significance of the results.

References

1. Gregg,C., Zhang,J., Weissbourd,B., Luo,S., Schroth,G.P., Haig,D. and Dulac,C. (2010) High-resolution analysis of parent-of-origin allelic expression in the mouse brain. *Science*, 329, 643–648.
2. DeVeale,B., van der Kooy,D. and Babak,T. (2012) Critical evaluation of imprinted gene expression by RNA-Seq: a new perspective. *PLoS Genet.*, 8, e1002600.
3. Bonthuis,P.J., Huang,W.-C., Stacher Hörndli,C.N., Ferris,E., Cheng,T. and Gregg,C. (2015) Noncanonical Genomic Imprinting Effects in Offspring. *Cell Rep*, 12, 979–991.
4. Babak,T. (2012) Identification of imprinted loci by transcriptome sequencing. *Methods Mol. Biol.*, 925, 79–88.
5. Babak,T., DeVeale,B., Tsang,E.K., Zhou,Y., Li,X., Smith,K.S., Kukurba,K.R., Zhang,R., Li,J.B., van der Kooy,D., et al. (2015) Genetic conflict reflected in tissue-specific maps of genomic imprinting in human and mouse. *Nat. Genet.*, 47, 544–549.
6. Perez,J.D., Rubinstein,N.D., Fernandez,D.E., Santoro,S.W., Needleman,L.A., Ho-Shing,O., Choi,J.J., Zirlinger,M., Chen,S.-K., Liu,J.S., et al. (2015) Quantitative and functional interrogation of parent-of-origin allelic expression biases in the brain. *Elife*, 4, e07860.
7. Hasin-Brumshtein,Y., Hormozdiari,F., Martin,L., van Nas,A., Eskin,E., Lusk,A.J. and Drake,T.A. (2014) Allele-specific expression and eQTL analysis in mouse adipose tissue. *BMC Genomics*, 15, 471.
8. Lorenc,A., Linnenbrink,M., Montero,I., Schilhabel,M.B. and Tautz,D. (2014) Genetic differentiation of hypothalamus parentally biased transcripts in populations of the house mouse implicate the Prader-Willi syndrome imprinted region as a possible source of behavioral divergence. *Mol. Biol. Evol.*, 31, 3240–3249.
9. Pirinen,M., Lappalainen,T., Zaitlen,N.A., GTEx Consortium, Dermitzakis,E.T., Donnelly,P., McCarthy,M.I. and Rivas,M.A. (2015) Assessing allele-specific expression across multiple tissues from RNA-seq read data. *Bioinformatics*, 31, 2497–2504.
10. Reynès,C., Kister,G., Rohmer,M., Bouschet,T., Varrault,A., Dubois,E., Rialle,S., Journot,L. and Sabatier,R. (2020) ISoLDE: a data-driven statistical method for the inference of allelic imbalance in datasets with reciprocal crosses. *Bioinformatics*, 36, 504–513.

Reviewer #3 (Remarks to the Author):

Martini et al. report the development of a novel application, brewerIX, for detecting bi-allelic transcription on the basis of SNVs detected in RNA-Seq data. They further focus the outputs on known imprinted regions and the X-chromosome, where biallelic expression is particularly interesting and relevant as it identifies loss of imprinting or escape from X-inactivation, both due

to epigenetic silencing where misregulation can have a detrimental impact on organismal health and other significant biological consequences. Martini et al convincingly demonstrate that brewerIX accurately detected LOI/X inactivation escape on samples where these effects were previously established and generated an easy-to-use package that requires minimal IT know-how and no programming expertise.

I believe there is a need for a tool like brewerIX and it could even seed a foundation for further development where additional features are added on later. The plots generated by the tool and presented in the paper offer a succinct and informative perspective, and I believe would be valuable to researchers studying processed where maintenance of imprinting and X-inactivation are relevant, and perhaps even other researchers looking for additional insights into their RNA-Seq data. I really like that there is a Mac/Linux package as this indeed a bottleneck that frequently cripples uptake of otherwise excellent informatics tools.

There are some issues, however, that prevented me from recommending publication. I summarize below.

1) I think the authors could do more optimization to gain insight into false discovery and sensitivity. It is nice that the settings decided on yield a relatively low single-digit false-positive rate, but the false-negative rate is concerningly high (well above 50%). Why is the tool missing so many expressed genes? I would expect the majority to have SNPs and only a small proportion to be genetically silent (cis-eQTL). I would recommend an ROC or precision-recall analysis following a titration of all parameters. This could also yield some insights where it could be possible to allow the end-user to dial in accepted FDR/sensitivity thresholds.

2) I like that the authors are using MAC as a parameter. Could this be expanded to "unique" reads (e.g. detected via UMIs or reads mapped to unique coordinates)? This has been in my past experience an exceptionally useful metric for distinguishing allelic expression from RNA-Seq PCR biases (e.g. SNPs inhibiting priming on one allele).

3) I worry that not enough explanation exists around how to interpret the data for end users. No detected SNVs in imprinted genes/X does not necessarily mean that imprinting/X-inactivation are present, as end-users might conclude. Monoallelic or allele-biased expression is often caused by other mechanisms and exists outside these regions. This caveat should be made clear.

4) My biggest disappointment is that the software didn't work for me. Install and launch was easy and the updates along the way useful indicators of progress, but for two separate it exited with an error about 40 minutes into analysis. I was aligning paired-end mouse RNAseq data from brain samples where the mouse were generated by crossing two well characterized inbred strains with lots of known SNPs. I attach some potentially useful tidbits from my log file below. I would highly recommend that if this is to be advertised as an easy to deploy software that it is also thoroughly tested by external users in novel environments.

```
[Sun, 12 Sep 2021 18:50:13 UTC] Mark duplicates
11:50:13.421 INFO NativeLibraryLoader - Loading libgkl_compression.dylib from
jar:file:/Users/defaultuser/.cache/brewerix/8/conda/envs/brewerix/share/picard-2.26.2-
0/picard.jar!/com/intel/gkl/native/libgkl_compression.dylib
[Sun Sep 12 11:50:13 PDT 2021] MarkDuplicates INPUT=[my_test_srr.1.bam]
OUTPUT=./tmpidwynhr7/my_test_srr.1.bam
METRICS_FILE=./tmpidwynhr7/my_test_srr.1_dupStats_matrix.txt
MAX_SEQUENCES_FOR_DISK_READ_ENDS_MAP=50000
MAX_FILE_HANDLES_FOR_READ_ENDS_MAP=8000 SORTING_COLLECTION_SIZE_RATIO=0.25
TAG_DUPLICATE_SET_MEMBERS=false REMOVE_SEQUENCING_DUPLICATES=false
TAGGING_POLICY=DontTag CLEAR_DT=true DUPLEX_UMI=false ADD_PG_TAG_TO_READS=true
REMOVE_DUPLICATES=false ASSUME_SORTED=false
DUPLICATE_SCORING_STRATEGY=SUM_OF_BASE_QUALITIES
PROGRAM_RECORD_ID=MarkDuplicates PROGRAM_GROUP_NAME=MarkDuplicates
READ_NAME_REGEX=<optimized capture of last three ':' separated fields as numeric values>
```

OPTICAL_DUPLICATE_PIXEL_DISTANCE=100 MAX_OPTICAL_DUPLICATE_SET_SIZE=300000
VERBOSITY=INFO QUIET=false VALIDATION_STRINGENCY=STRICT COMPRESSION_LEVEL=5
MAX_RECORDS_IN_RAM=500000 CREATE_INDEX=false CREATE_MD5_FILE=false
GA4GH_CLIENT_SECRETS=client_secrets.json USE_JDK_DEFLATER=false
USE_JDK_INFLATER=false
[Sun Sep 12 11:50:13 PDT 2021] Executing as defaultuser@defaultusers-MacBook-Pro.local on
Mac OS X 10.14.6 x86_64; OpenJDK 64-Bit Server VM 1.8.0_302-b08; Deflater: Intel; Inflater:
Intel; Provider GCS is not available; Picard version: 2.26.2
INFO 2021-09-12 11:50:13 MarkDuplicates Start of doWork freeMemory: 501735632;
totalMemory: 514850816; maxMemory: 1908932608
INFO 2021-09-12 11:50:13 MarkDuplicates Reading input file and constructing read end
information.
INFO 2021-09-12 11:50:13 MarkDuplicates Will retain up to 6916422 data points before spilling to
disk.
WARNING 2021-09-12 11:50:13 AbstractOpticalDuplicateFinderCommandLineProgram A field field
parsed out of a read name was expected to contain an integer and did not. Read name:
my_test_srr.1.35200113. Cause: String 'my_test_srr.1.35200113' did not start with a parsable
number.
INFO 2021-09-12 11:50:19 MarkDuplicates Read 1,000,000 records. Elapsed time: 00:00:05s.
Time for last 1,000,000: 5s. Last read position: X:10,718,496
INFO 2021-09-12 11:50:19 MarkDuplicates Tracking 34883 as yet unmatched pairs. 1783 records
in RAM.
INFO 2021-09-12 11:50:22 MarkDuplicates Read 2,000,000 records. Elapsed time: 00:00:09s.
Time for last 1,000,000: 3s. Last read position: X:134,805,101
INFO 2021-09-12 11:50:22 MarkDuplicates Tracking 73358 as yet unmatched pairs. 4713 records
in RAM.
INFO 2021-09-12 11:50:24 MarkDuplicates Read 2530030 records. 89655 pairs never matched.
INFO 2021-09-12 11:50:25 MarkDuplicates After buildSortedReadEndLists freeMemory:
725205952; totalMemory: 1014497280; maxMemory: 1908932608
INFO 2021-09-12 11:50:25 MarkDuplicates Will retain up to 59654144 duplicate indices before
spilling to disk.
INFO 2021-09-12 11:50:25 MarkDuplicates Traversing read pair information and detecting
duplicates.
INFO 2021-09-12 11:50:26 OpticalDuplicateFinder Large duplicate set. size = 1112
INFO 2021-09-12 11:50:26 OpticalDuplicateFinder compared 1,000 ReadEnds to others. Elapsed
time: 00:00:00s. Time for last 1,000: 0s. Last read position: -1:-1
INFO 2021-09-12 11:50:26 MarkDuplicates Traversing fragment information and detecting
duplicates.
INFO 2021-09-12 11:50:27 MarkDuplicates Sorting list of duplicate records.
INFO 2021-09-12 11:50:27 MarkDuplicates After generateDuplicateIndexes freeMemory:
1323528728; totalMemory: 1816657920; maxMemory: 1908932608
INFO 2021-09-12 11:50:27 MarkDuplicates Marking 952513 records as duplicates.
INFO 2021-09-12 11:50:27 MarkDuplicates Found 0 optical duplicate clusters.
INFO 2021-09-12 11:50:27 MarkDuplicates Reads are assumed to be ordered by: coordinate

A fatal error has been detected by the Java Runtime Environment:

SIGSEGV (0xb) at pc=0x000000010b03fea7, pid=30163, tid=0x0000000000002203

JRE version: OpenJDK Runtime Environment (Zulu 8.56.0.21-CA-macosx) (8.0_302-b08) (build
1.8.0_302-b08)
Java VM: OpenJDK 64-Bit Server VM (25.302-b08 mixed mode bsd-amd64 compressed oops)
Problematic frame:
C [libgkl_compression2530869309439406063.dylib+0x6ea7] deflate_medium+0x867

Failed to write core dump. Core dumps have been disabled. To enable core dumping, try "ulimit
-c unlimited" before starting Java again

An error report file with more information is saved as:

```
# /Users/defaultuser/Desktop/untitled folder/hs_err_pid30163.log
#
# If you would like to submit a bug report, please visit:
# http://www.azul.com/support/
# The crash happened outside the Java Virtual Machine in native code.
# See problematic frame for where to report the bug.
#
/Users/defaultuser/.cache/brewerix/8/conda/envs/brewerix/bin/picard: line 66: 30163 Abort trap:
6 /Users/defaultuser/.cache/brewerix/8/conda/envs/brewerix/bin/java -Xms512m -Xmx2g -jar
/Users/defaultuser/.cache/brewerix/8/conda/envs/brewerix/share/picard-2.26.2-0/picard.jar
MarkDuplicates "I=" "my_test_srr.1.bam" "O=" "./tmpidwynhr7/my_test_srr.1.bam" "M="
"./tmpidwynhr7/my_test_srr.1_dupStats_matrix.txt"
Workflow guess LOI version: 0.11.0
Traceback (most recent call last):
File "/Users/defaultuser/.cache/brewerix/8/conda/envs/brewerix/bin/guess-LOI", line 8, in
<module>
sys.exit(guess_loi())
File "/Users/defaultuser/.cache/brewerix/8/conda/envs/brewerix/lib/python3.7/site-
packages/workflow/guess_loi/guess_loi.py", line 47, in guess_loi
guess_loi_from_fqs(args)
File "/Users/defaultuser/.cache/brewerix/8/conda/envs/brewerix/lib/python3.7/site-
packages/workflow/guess_loi/guess_loi.py", line 91, in guess_loi_from_fqs
bams = mark_duplicates(bams, samples, p, clean=args.clean)
File "/Users/defaultuser/.cache/brewerix/8/conda/envs/brewerix/lib/python3.7/site-
packages/workflow/guess_loi/mark_duplicates.py", line 25, in mark_duplicates
"M=", tmp_matrix])
File "/Users/defaultuser/.cache/brewerix/8/conda/envs/brewerix/lib/python3.7/subprocess.py",
line 363, in check_call
raise CalledProcessError(retcode, cmd)
subprocess.CalledProcessError: Command '['picard', 'MarkDuplicates', 'I=', 'my_test_srr.1.bam',
'O=', './tmpidwynhr7/my_test_srr.1.bam', 'M=',
'./tmpidwynhr7/my_test_srr.1_dupStats_matrix.txt']' returned non-zero exit status 134.
Command failed with code: 1
```

Response letter

Dear Reviewers,

We thank you for your constructive comments that helped us in significantly improving our manuscript.

We appreciated your expert opinion and we edited the manuscript to include all the limitations of our approach, in comparison to relevant studies. We have also written a new paragraph about the interpretation of the results obtained with our tool, as suggested.

We improved the code setting with additional environment variables that help us in identifying potential errors and we had also already implemented a bug tracker. We believe all these actions will help us solve future errors and issues experienced by users.

Finally, we have performed the precision-recall analysis (new Figure 4) showing that in a range of parameters (including our default parameters) the precision and recall are in an acceptable range, a very important information for future users of our tool.

We believe that our manuscript and tool are now more solid and comprehensive, thanks to your suggestions and comments.

Best wishes,
Graziano Martello and Chiara Romualdi

Point-by-point response

Reviewer #1 (Remarks to the Author):

The manuscript presents a software tool, BrewerIX, for the study of loss-of-imprinting (LOI) and X-inactivation from RNA-seq data coupled with genotype information, designed to serve as a standardised and easy to use pipeline for such analyses. After the first round of review, the manuscript was transferred to Communications Biology and I am now commenting on this manuscript as a new reviewer. While I have not seen the original manuscript, based not the rebuttal and the new version of the manuscript, I find the authors have done comprehensive work in revising the text and analyses. Particularly, the rationale for the development of the tool is now clearly stated.

While I have no major comments, I ask the authors to address a few points:

1 - My biggest concern is the suitability of their software tool for the analysis of X-inactivation using human bulk RNA-seq. In the introduction, the authors mention the "random epigenetic silencing mechanism called X chromosome inactivation (XCI)", yet I failed to find any discussion on how the randomness of the mechanism potentially impacts the use of ASE for the analysis of X-inactivation. In normal human female tissue samples, X-inactivation is indeed random, i.e. about 50% of the cells express the paternal and 50% the maternal copy of X, and hence in bulk RNA-seq most of the X-chromosomal sites are expressed bi-allelically irrespective of their X-inactivation status (i.e. escapee or silenced). In single cell RNA-seq the randomness of X-inactivation is obviously not an issue, and e.g. in mouse X-inactivation is imprinted with preferential inactivation of the paternal X, and therefore the suggested tool should work fine for the inference of X-inactivation in these types of data.

While it appears that the authors do not present any X-inactivation results from human bulk RNA-seq data, it should nevertheless be made very clear in the manuscript that such a limitation in the application of the tool exists.

We agree with the reviewer and we edited the Introduction (line 53) and the Discussion (lines 313-317) accordingly. Indeed, as pointed out, we do not present any X-inactivation results from bulk RNA-seq.

2 - "the genes on the X chromosome are expressed mono-allelically" In female samples, this mono-allelic expression does apply to the majority of the X chromosome genes, but a considerable proportion (~20-25%) of X-linked genes escape from X-inactivation in humans and are expressed bi-allelically. Additionally, in relation to my first comment, in bulk RNA-seq it is unlikely that a large number of X chromosome genes are mono-allelic in a human female sample, unless there is considerable skewing of X-inactivation.

We edited the text as suggested (line 51).

3 - "X-linked and imprinting diseases are the most common congenital human disorders because loss-of-function mutations in the single expressed allele will not be buffered by the second silenced allele" While this statement is true for imprinted genes, again I ask the authors to specify how this holds for the X chromosome. Given the randomness of X-inactivation, females are most often protected from X-linked of mutations.

We edited the text as suggested (line 53).

4 - While I appreciate the idea of the tool be as a low threshold option for LOI and chrX analyses, I would call for a more extensive discussion on the potential limitations there are in such an automated pipeline vs. customised settings. These limitations may include, e.g., reference allele bias in RNA-seq alignment, handling of RNA-seq reads that overlap multiple SNVs, quality control of heterozygous sites for ASE, poor mappability of some genomic regions, and handling of duplicated RNA-seq reads, that I did not see were specifically discussed or included in the methods.

Our goal was to make easily accessible complicated bioinformatic pipelines for the sake of data reproducibility. We do agree with the referee that the inference of gene imprinting and X-chr status is a challenging task and that

there are several limitations when using RNA-seq data. On the other hand our tool is designed to be fully customizable by advanced users with specific python packages in order to bypass possible limitations.

Following the reviewer suggestion, we expand the text including a dedicated section titled "*Limitations and guidelines for the interpretation of data*" highlighting the possible limits of this approach such as the reference bias mappability and the poor mappability of some genomic regions. The handling of duplicated reads and of reads that overlap 2 SNVs are already taken into account by GATK itself, the software used to call ASE counts. QC of heterozygous sites is again something that GATK should take care of excluding reads with poor quality from the final counts. We made all these issues clearer in the text.

Reviewer #2 (Remarks to the Author):

Assessing allelic imbalance in RNAseq data is not easy. The caveats of the different approaches were best illustrated by the controversy about the number and identity of imprinted genes that followed the publication of a manuscript by Gregg and colleagues a decade ago (1). The Babak, Dulac, and Gregg groups then proposed modifications of the existing statistical methods that led to contradictory conclusions (2–6). Other groups also proposed different parametric methods (7–9). More recently, a non-parametric method that calls both bi-allelic and biased expression was proposed (10). Despite sophisticated statistical methods, the use of replicated RNAseq data, and reciprocal crosses of mouse strains, the conflicting results illustrated the difficulty of the task and the caution that should be taken to call a gene expression biased.

Considering the existing literature, the approach proposed by Martini and colleagues is naive and simplistic. It is just not possible to reliably call bi-allelic gene expression by only filtering allelic ratios. The second calling method is to use a binomial test. Given the literature above, this is again a simplistic approach unable to deal with the intrinsic complexity of the task. Furthermore, the manuscript offers no detail on how the test is performed exactly and does not mention multiple testing correction; the reference given to support the use of the test (#29) is a review article on X inactivation, which does not describe binomial test whatsoever.

In their rebuttal to reviewer 2, the authors emphasized the fact that BrewerIX is easy to install, fast, and does not require computing skills. This is likely true, but clearly at the expense of accuracy and significance of the results.

References

1. Gregg,C., Zhang,J., Weissbourd,B., Luo,S., Schroth,G.P., Haig,D. and Dulac,C. (2010) **High-resolution analysis of parent-of-origin allelic expression in the mouse brain**. *Science*, 329, 643–648.
2. DeVeale,B., van der Kooy,D. and Babak,T. (2012) **Critical evaluation of imprinted gene expression by RNA-Seq: a new perspective**. *PLoS Genet.*, 8, e1002600.
3. Bonthuis,P.J., Huang,W.-C., Stacher Hörndli,C.N., Ferris,E., Cheng,T. and Gregg,C. (2015) Noncanonical Genomic Imprinting Effects in Offspring. *Cell Rep*, 12, 979–991.
4. Babak,T. (2012) **Identification of imprinted loci by transcriptome sequencing**. *Methods Mol. Biol.*, 925, 79–88.
5. Babak,T., DeVeale,B., Tsang,E.K., Zhou,Y., Li,X., Smith,K.S., Kukurba,K.R., Zhang,R., Li,J.B., van der Kooy,D., et al. (2015) Genetic conflict reflected in tissue-specific maps of genomic imprinting in human and mouse. *Nat. Genet.*, 47, 544–549.
6. Perez,J.D., Rubinstein,N.D., Fernandez,D.E., Santoro,S.W., Needleman,L.A., Ho-Shing,O., Choi,J.J., Zirlinger,M., Chen,S.-K., Liu,J.S., et al. (2015) **Quantitative and functional interrogation of parent-of-origin allelic expression biases in the brain**. *Elife*, 4, e07860.
7. Hasin-Brumshtein,Y., Hormozdiari,F., Martin,L., van Nas,A., Eskin,E., Lusis,A.J. and Drake,T.A. (2014) Allele-specific expression and eQTL analysis in mouse adipose tissue. *BMC Genomics*, 15, 471.
8. Lorenc,A., Linnenbrink,M., Montero,I., Schilhabel,M.B. and Tautz,D. (2014) Genetic differentiation of hypothalamus parentally biased transcripts in populations of the house mouse implicate the Prader-Willi syndrome imprinted region as a possible source of behavioral divergence. *Mol. Biol. Evol.*, 31, 3240–3249.
9. Pirinen,M., Lappalainen,T., Zaitlen,N.A., GTEx Consortium, Dermitzakis,E.T., Donnelly,P., McCarthy,M.I. and Rivas,M.A. (2015) **Assessing allele-specific expression across multiple tissues from RNA-seq read data**. *Bioinformatics*, 31, 2497–2504.

10. Reynès,C., Kister,G., Rohmer,M., Bouschet,T., Varrault,A., Dubois,E., Rialle,S., Journot,L. and Sabatier,R. (2020) ISoLDE: a data-driven statistical method for the inference of allelic imbalance in datasets with reciprocal crosses. *Bioinformatics*, 36, 504–513.

We take note of the skepticism of the Referee regarding the possibility of investigating LOI through RNA-seq data, and we agree with him that it is a challenging task. However the Referee has to consider that there is a large scientific literature on the use of RNA-seq to study the loss of imprinting in many physiological states and that these studies provided extensive validation in support of their findings (Pastor et al, *Cell Stem Cell* 2016; Santini et al, *Nature Communications* 2021; Choi et al, *Nature* 2017 and several others). Although naive and simplistic with respect to other more sophisticated approaches, our method has the advantage to guarantee reproducibility, it is flexible and customizable, and more importantly it is characterised by a high precision-recall rate as reported in this new revised version. We have also provided in our work validation of our results, either empirical (through the replicability of the results obtained by other studies) and experimental through sanger and whole exome sequence data. Thus, beyond its simplicity, BrewerIX provides reliable predictions that of course have to be further validated. Multiple testing in this context, as in all the other omic fields, is certainly a problem as with the increasing number of tests the false positives tend to increase, however we set our thresholds in order to control the False Discovery Rate (FDR) as reported in Figure 1b. Regarding the p-value obtained with the binomial test we give the possibility to the user to set a stringent cut-off.

We apologise for the previous reference #29, which did not describe a binomial test. That was a mistake made during preparation of the manuscript. The correct references have been added (currently #22 and #23).

Finally, recognizing that the Referee mentioned important alternative statistical approaches in the field, we included these references in the new revised version and we expanded the Introduction and Discussion to comment on possible limitations and difficulties linked to the use of RNA-seq for LOI inference.

Reviewer #3 (Remarks to the Author):

Martini et al. report the development of a novel application, brewerIX, for detecting bi-allelic transcription on the basis of SNVs detected in RNA-Seq data. They further focus the outputs on known imprinted regions and the X-chromosome, where biallelic expression is particularly interesting and relevant as it identifies loss of imprinting or escape from X-inactivation, both due to epigenetic silencing where misregulation can have a detrimental impact on organismal health and other significant biological consequences. **Martini et al convincingly demonstrate that brewerIX accurately detected LOI/X inactivation escape on samples where these effects were previously established and generated an easy-to-use package that requires minimal IT know-how and no programming expertise.**

I believe there is a need for a tool like brewerIX and it could even seed a foundation for further development where additional features are added on later. The plots generated by the tool and presented in the paper offer a succinct and informative perspective, and I believe would be valuable to researchers studying processed where maintenance of imprinting and X-inactivation are relevant, and perhaps even other researchers looking for additional insights into their RNA-Seq data. I really like that there is a Mac/Linux package as this indeed a bottleneck that frequently cripples uptake of otherwise excellent informatics tools.

We thank the reviewers for the positive comments.

There are some issues, however, that prevented me from recommending publication. I summarize below.

1- I think the authors could do **more optimization to gain insight into false discovery and sensitivity.** It is nice that the settings decided on yield a relatively low single-digit false-positive rate, but the false-negative rate is concerningly high (well above 50%). Why is the tool missing so many expressed genes? I would expect the majority to have SNPs and only a small proportion to be genetically silent (cis-eQTL). I would **recommend an ROC or**

precision-recall analysis following a titration of all parameters. This could also yield some insights where it could be possible to **allow the end-user to dial in accepted FDR/sensitivity thresholds.**

Following the Referee suggestion we reported in the new Figure 4 the precision/recall analysis based on the set of genes detected by RNA-seq data analyses and validated by Sanger sequencing both i) in our study and ii) in the study by Santini and colleagues (Nature Communications 2021).

We pooled together all the experimental validations coming from different samples obtaining a total of 307 validated SNP of which 49 are true positives (i.e. validated bi allelic) and 258 are True Negatives (i.e. validated mono-allelic expression).

From the new Figure 4 we observed that the precision has an increasing trend from lower to higher overall depth thresholds. For Allelic Ratios (AR) between 0.1 and 0.3 we obtained a precision greater than 80% for overall depth cut-off greater or equal to 15 reads. The recall was above 60% for values equal to, or lower than 20 for the overall depth.

We also tested the performance of the binomial distribution, rather than using AR, and found a Precision above 90% for p-values equal to or below 0.05, regardless of the overall depth, while the Recall was affected by the p-value, remaining above 60% only in the case of a p-value equal to 0.05 and an overall depth equal to or below 15.

This analysis, performed on a set of experimentally validated genes, confirms that our default parameter of overall depth=20, AR=2 (thus a minor allele count = 4) gives the best trade-off for Precision and Recall.

Moreover, using a binomial distribution with a p-value<=0.05 and an overall depth=15 gives even higher precision with an acceptable Recall.

NEW FIGURE 4:

2- I like that the authors are using MAC as a parameter. **Could this be expanded to "unique" reads (e.g. detected via UMIs or reads mapped to unique coordinates)?** This has been in my past experience an exceptionally useful metric for distinguishing allelic expression from RNA-Seq PCR biases (e.g. SNPs inhibiting priming on one allele).

The pipeline is set to remove coordinate-based reads duplicates. Nevertheless the tool can be used starting from bam files created by the user where a UMI based duplicated flag has been carried out. Moreover all python libraries are made freely available and an advanced user can fully customize the implementation while keeping the same end visualization style.

3- I worry that **not enough explanation exists around how to interpret the data** for end users. **No detected SNVs in imprinted genes/X does not necessarily mean that imprinting/X-inactivation are present**, as end-users might conclude. **Monoallelic or allele-biased expression is often caused by other mechanisms and exists outside these regions.** This caveat should be made clear.

We thank the reviewer for the constructive comment. We have added a new section in which we explain in detail how to interpret the results generated by our tool. See lines 299-340.

4) My **biggest dissapointment is** that the software didn't work for me. Install and launch was easy and the updates along the way useful indicators of progress, but for two separate it exited with an error about 40 minutes into analysis. I was aligning paired-end mouse RNAseq data from brain samples where the mouse were generated by crossing two well characterized inbred strains with lots of known SNPs. I attach some potentially useful tidbits from my log file below. I would highly recommend that if this is to be advertised as an easy to deploy software that it is also thoroughly tested by external users in novel environments.

We are sorry for this inconvenience and we thank the referee for the note. We improved the code setting additional environment variables that help us in identifying the errors. For eventuality like this, we had already implemented a bug tracker at <https://github.com/Romualdi-Lab/brewerix-cli> that helps us to improve and update the software. Nevertheless, thanks to the reviewer we improved the error message in BrewerIX user interface that now includes a button that brings the user to the bug report tracker system. Furthermore, we made available the link to the bug report system through the website, under the FAQs section.

REVIEWERS' COMMENTS:

Reviewer #1 (Remarks to the Author):

The authors have addressed by concerns in a sufficient manner. I particularly appreciate the new section on the limitations and guidelines for data interpretation. I have no further comments.

Reviewer #3 (Remarks to the Author):

The authors have satisfactorily addressed my concerns and I recommend publication of the revised manuscript.